# Realizing high-ranged thermoelectric performance in PbSnS$_2$ crystals

Shaoping Zhan[1], Tao Hong[1], Bingchao Qin [1], Yingcai Zhu [1], Xiang Feng[1], Lizhong Su[1], Haonan Shi[1], Hao Liang[2], Qianfan Zhang[1], Xiang Gao[3], Zhen-Hua Ge[2], Lei Zheng[1], Dongyang Wang [4] ✉ & Li-Dong Zhao [1,5] ✉

Great progress has been achieved in p-type SnS thermoelectric compound recently, while the stagnation of the n-type counterpart hinders the construction of thermoelectric devices. Herein, n-type sulfide PbSnS$_2$ with iso-structural to SnS is obtained through Pb alloying and achieves a maximum $ZT$ of ~1.2 and an average $ZT$ of ~0.75 within 300–773 K, which originates from enhanced power factor and intrinsically ultralow thermal conductivity. Combining the optimized carrier concentration by Cl doping and enlarged Seebeck coefficient through activating multiple conduction bands evolutions with temperature, favorable power factors are maintained. Besides, the electron doping stabilizes the phase of PbSnS$_2$ and the complex-crystal-structure induced strong anharmonicity results in ultralow lattice thermal conductivity. Moreover, a maximum power generation efficiency of ~2.7% can be acquired in a single-leg device. Our study develops a n-type sulfide PbSnS$_2$ with high performance, which is a potential candidate to match the excellent p-type SnS.

The thorny problems of energy shortage and environmental pollution have been challenging the sustainable development. Thus, thermoelectric materials have gained increasing attention because they provide an alternative way of energy utilization by realizing the direct and reversible conversion between heat and electricity[1–3]. The efficiency of thermoelectric devices depends on the material's dimensionless figure of merit, $ZT = S^2\sigma T/(\kappa_{lat} + \kappa_{ele})$, where $S$, $\sigma$, $S^2\sigma$, $T$, $\kappa_{lat}$ and $\kappa_{ele}$ represent Seebeck coefficient, electrical conductivity, power factor, absolute temperature, lattice thermal conductivity and electronic thermal conductivity, respectively. Numerous strategies have emerged to boost thermoelectric performance after decades of efforts, including enhancing Seebeck coefficient through distortion of the density of states[4], band convergence[5–7] and electron energy barrier filtering[8], optimizing carrier concentration and maintaining high carrier mobility to improve electrical conductivity[9–12], and designing all-scale hierarchical architectures to lower lattice thermal conductivity[13–15]. The potential of thermoelectric materials can be further motivated through applying these classical strategies synergistically.

The ideal thermoelectric devices for large-scale and high-efficiency applications prefer homojunction structures in order to avoid lattice mismatch incompatibility and harmful band misalignment. Therefore, the development of similar-system and performance matched n-type and p-type thermoelectric materials is particularly important. Recently, the low cost and environmentally friendly p-type SnS has been greatly developed and a high average $ZT$ of 1.25 was reported in SnS$_{0.91}$Se$_{0.09}$ crystals[16], mainly due to the synergistically optimization of carrier effective mass and mobility through dynamic valence bands interplay. However, the development of n-type SnS is far lagging behind due to the great difficulties and challenges on achieving effective electron doping to optimize the poor n-type electrical transports.

[1]School of Materials Science and Engineering, Beihang University, Beijing 100191, China. [2]Faculty of Materials Science and Engineering, Kunming University of Science and Technology, Kunming 650093, China. [3]Center for High Pressure Science and Technology Advanced Research (HPSTAR), Beijing 100094, China. [4]Henan Key Laboratory of Diamond Optoelectronic Materials and Devices, Key Laboratory of Material Physics, Ministry of Education, School of Physics, Zhengzhou University, Zhengzhou 450052, China. [5]Key Laboratory of Intelligent Sensing Materials and Chip Integration Technology of Zhejiang Province (2021E10022), Hangzhou Innovation Institute of Beihang University, Hangzhou 310051, China. ✉e-mail: wangdongyang@buaa.edu.cn; zhaolidong@buaa.edu.cn

To date, several attempts to realize n-type SnS have been conducted, including aliovalent cation doping (Nd, Sb and Bi)[17–19], deviating from stoichiometric ratio to compensate for Sn vacancies[20], and aliovalent anion doping (Cl and Br)[21–24]. However, the electron carrier concentration obtained through the above approaches is far below the optimal carrier concentration level required for high-performance thermoelectric materials[25]. The remaining one glimmer of hope is that isovalent cation Pb alloying favors electron generation through suppressing Sn vacancies and forming interstitials $Sn_i$ and $Pb_i$[26,27]. Meanwhile, researchers have concluded that the solid solubility of Pb element in SnS is slightly more than 50%[28,29], and the solid solubility of Pb in Sn positions will significantly favor the effective electron doping in SnS[26]. By using Pb alloy at the Sn sites in SnS, another promising sulfide compound, $PbSnS_2$, was obtained to approach promising n-type thermoelectric performance. Surprisingly, similar to SnS and SnSe[30–33], $PbSnS_2$ is composed of low-cost elements, possesses the layered and orthorhombic crystal structure with *Pnma* space group, the wide bandgap ~0.9–1.12 eV, and the intrinsically low thermal conductivity[34,35]. All of these completely meet the selection rules of potentially high-performance thermoelectric materials[36]. However, apart from the theoretical calculations[37], $PbSnS_2$-based thermoelectrics have few experimental achievements due to its ultralow carrier concentration and poor electrical conductivity[26], making it rather urgent to develop and establish the proper n-type transports for $PbSnS_2$ as the counterpart of p-type SnS.

Strategies of growing crystals and utilizing the unique layered crystal structure have been developed to optimize the electrical performance in both SnSe and SnS crystals[2,5,9,16,32,38–40]. Herein, we successfully synthesized $PbSnS_2$ crystals by a modified temperature gradient method and investigated their thermoelectric transport performance in detail. Compared with the in-plane direction, our results show that higher thermoelectric performance is achieved along the out-of-plane direction due to the ultralow thermal conductivity coming from strong interlayer phonon scattering and superior power factor at higher temperatures (>600 K) due to increased interlayer charge density, which is similar to the anisotropy of n-type SnSe[9,38]. The undoped $PbSnS_2$ crystal shows typical semiconductor transport behavior with rising temperature due to the low carrier concentration. And it might be partially decomposed into PbS and SnS at ~623 K, while PbS as a high thermal conductivity phase is harmful to thermoelectric transport[41]. To improve the poor electrical transport performance of the pristine $PbSnS_2$ crystals, the halogen element Cl was selected as electron dopant due to its similar ionic radius to S. Surprisingly, Cl doping enhances the phase stability of $PbSnS_2$ crystals compared with the undoped sample, and their lattice thermal conductivity follows the Umklapp mechanism to remain the low values[42]. The aliovalent substitution of Cl at S sites significantly optimizes the carrier concentration and thus increases the electrical conductivity. Also, the effective doping pushes Fermi level deeper and activates multiple conduction bands with increasing temperature, maintaining the large Seebeck coefficient. Resultantly, the power factor, lattice thermal conductivity and *ZT* values of the optimal Cl doped $PbSnS_2$ crystal are ~4.26 μW cm$^{-1}$ K$^{-2}$, ~0.25 W m$^{-1}$ K$^{-1}$ and ~1.2 at 773 K, respectively, showing tremendous optimization after Cl doping. Additionally, certain performance of the output power and energy conversion efficiency is acquired in a single-leg device based on our high-performance $PbSnS_2$ crystals. This study demonstrates that Cl doped $PbSnS_2$ crystals can be the promising n-type thermoelectric candidates. Our attempts on single-leg devices also develop a potential n-type counterpart of p-type SnS to form low-cost and full-scale thermoelectric devices.

## Results

### Crystal structure characterization

To clarify the crystal structure of $PbSnS_2$, X-ray diffraction (XRD) analysis and scanning transmission electron microscope (STEM) characterization were performed. A typical $PbSnS_2$ crystal prepared by the modified temperature gradient method and the measurement samples along different crystalline directions are shown in Fig. 1a. Powder X-ray diffraction patterns for undoped $PbSnS_2$ (carrier concentration is $3.6 \times 10^{11}$ cm$^{-3}$) and Cl doped $PbSnS_2$ (carrier concentrations are $0.8 \times 10^{19}$, $1.3 \times 10^{19}$ and $1.7 \times 10^{19}$ cm$^{-3}$, respectively) prove that the single phases were synthesized (Supplementary Fig. 1). While the X-ray diffraction patterns for corresponding cleavage plane present only two diffraction peaks located at 31.5° and 65.5°, which can be assigned to the (004) and (008) crystal planes within the angle range of scanning, indicating the high quality of $PbSnS_2$ crystals (Fig. 1b). Scanning transmission electron microscopy-high angle annular dark field (STEM-HAADF) was adopted to observe the microstructure of $PbSnS_2$ crystals along the zone axes of [100], [110] and [001], corresponding to Fig. 1c, e, f, respectively. Furthermore, combined with the annular bright field (ABF) image along the zone axis of [100] (Fig. 1d), which was obtained with the corresponding HAADF image at the same time, the atom locations of heavy elements Pb or Sn and light element S atoms can be clearly seen. One can see that the zig-zag accordion-like structure along intralayers and weak bonding along interlayers form a natural layered structure in $PbSnS_2$, hinting that it is an intrinsic low thermal conductivity material with strong anisotropy and anharmonicity. These observations indicate that we have obtained high-quality $PbSnS_2$ crystals, which have layered orthorhombic crystal structure with *Pnma* space group like SnSe and SnS systems[9,16,32]. As for the relative positions of Pb and Sn in $PbSnS_2$, four models, named as Models 0, A, B and C (Supplementary Fig. 2), have been summarized in the literature, and their structures have been characterized in detail by high resolution electron microscopy[43]. Herein, the calculated total energy of these four models suggests that Model A is the most favorable configuration, and we carried out the later Rietveld refinement based on Model A due to the lowest energy (Supplementary Fig. 3a). Meanwhile, Supplementary Fig. 3b–d demonstrate that the diffraction peaks and the local environment for Pb or Sn based on Model A are in good agreement with the experimental results, indicating the rationality of Model A, which is also adopted by Kanatzidis et al.[44].

### Electrical transport properties

The temperature-dependent electrical transport performance of undoped and Cl doped crystals along the out-of-plane direction is shown in Fig. 2. The undoped $PbSnS_2$ crystal exhibits the insulator-like behavior at room temperature due to the ultralow carrier concentration (~$3.6 \times 10^{11}$ cm$^{-3}$), and its electrical transport performance cannot be estimated accurately within the detection ability of the instrument (Fig. 2a). Specifically, the electrical conductivity of undoped sample increases from ~0.001 to ~0.86 S cm$^{-1}$ in the temperature range of 423–823 K, indicating the process of thermal excitation and the semiconductor transport behavior. In order to optimize the carrier concentration and electrical transport performance of $PbSnS_2$, the halogen element Cl was chosen to substitute S sites due to their similar ionic radius[45]. It is worth noting that the electrical conductivity of $PbSnS_2$ crystals have been significantly boosted after Cl doping, showing a typical metal-like transport behavior. It appears that the electrical conductivity increases with the improvement of carrier concentration. Especially in the optimal Cl doped $PbSnS_2$ crystal, the electrical conductivity can reach ~140 S cm$^{-1}$ at 300 K due to a high carrier concentration of ~$1.7 \times 10^{19}$ cm$^{-3}$, indicating that Cl is an effective n-type dopant to optimize the carrier concentration in $PbSnS_2$. Figure 2b shows the temperature-dependent Seebeck coefficient. As we can see, negative Seebeck coefficient corresponds to n-type conduction, which is consistent with experimental results in the literature[26]. According to the calculated defect formation energy, vacancies of Sn and Pb would be suppressed under S-poor condition[26,27]. Also, the lattice parameter of $PbSnS_2$ along the out-of-plane direction increases by ~0.24 Å compared with SnS[30] because of

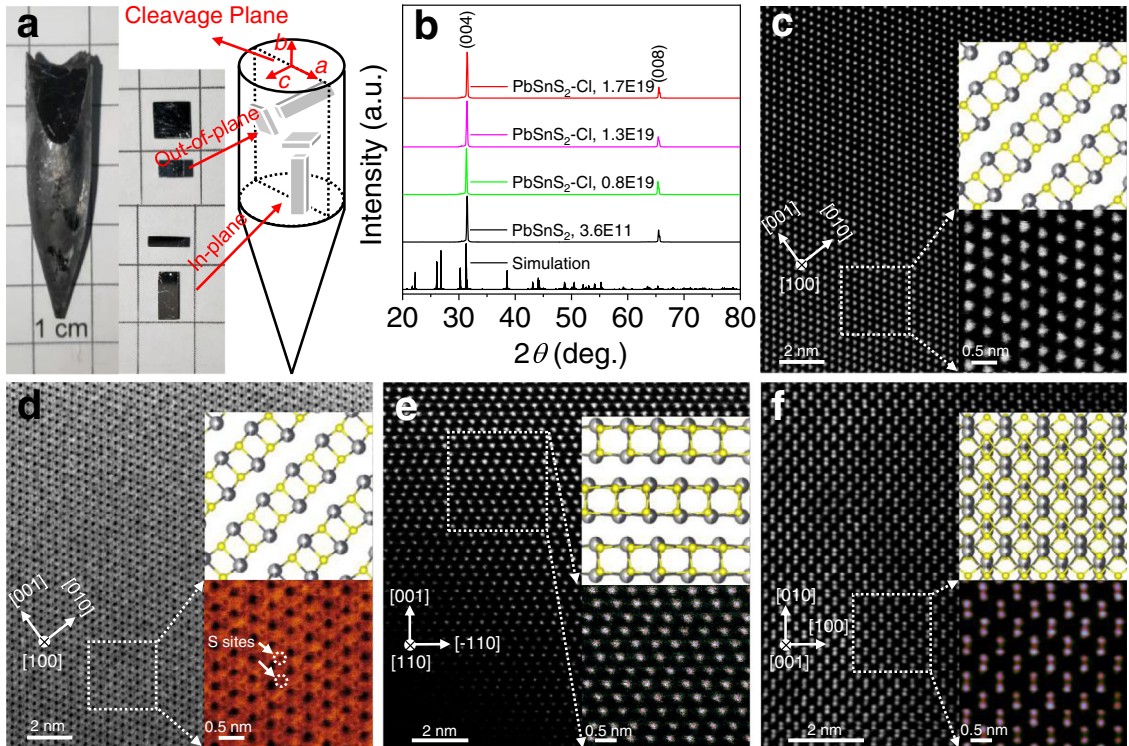

**Fig. 1 | Crystal structure characterization of PbSnS₂ at room temperature. a** A typical PbSnS₂ crystal cleaved along the (001) plane (left), and the schematic diagram of how the sample is cut along the out-of-plane and in-plane directions for thermoelectric performance measurements (right). **b** XRD patterns of undoped and Cl doped PbSnS₂ along the cleavage plane and corresponding simulation pattern. The HAADF images along the zone axes of (**c**) [100], (**e**) [110] and (**f**) [001]. **d** The ABF image along the zone axis [100]. Local magnification and crystal structure model are presented at the right of the corresponding STEM image, where the black and gray mixed spheres represent Pb or Sn atoms, and the yellow spheres represent S atoms.

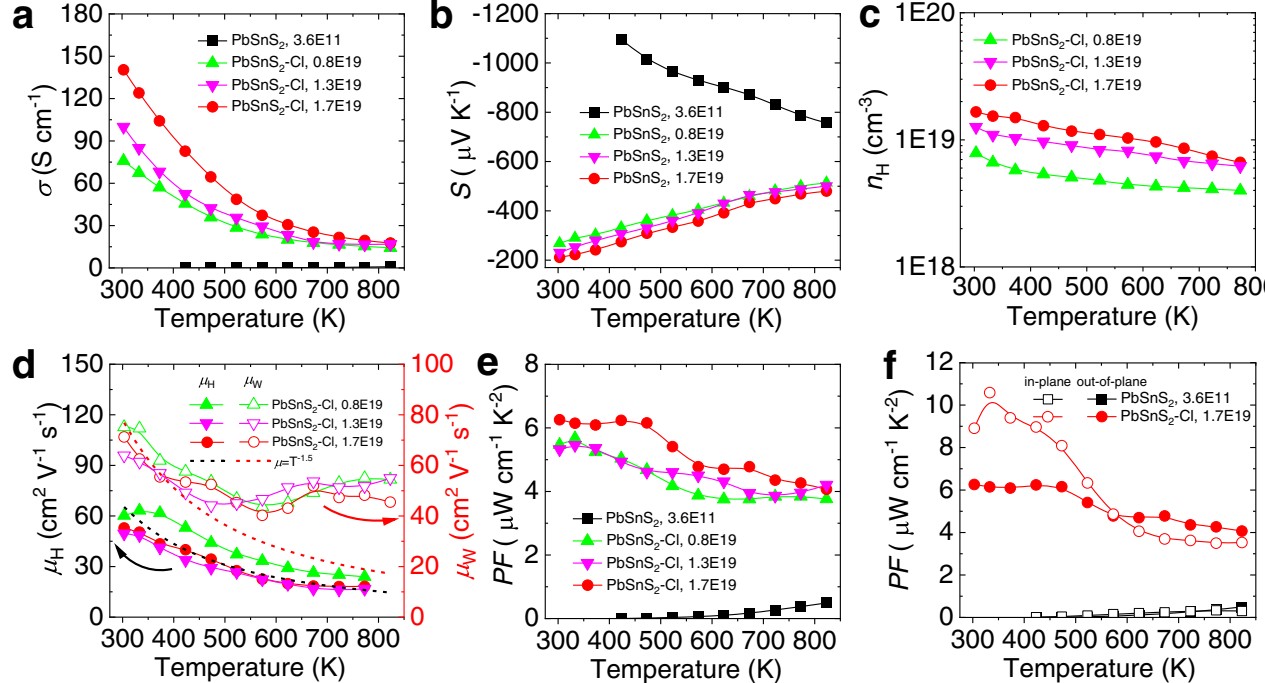

**Fig. 2 | Temperature-dependent electrical transport performance of undoped and Cl doped PbSnS₂ along the out-of-plane direction. a** Electrical conductivity. **b** Seebeck coefficient. **c** Carrier concentration. **d** Carrier mobility and weighted mobility. **e** Power factor. **f** Power factor comparisons.

the enlarged interlayer distance by larger $Pb^{2+}$ ions substitution, which is conducive to the formation of interstitials $Sn_i$ and $Pb_i$[27]. These interstitials contribute a lot of electrons due to $Sn^0/Pb^0 = Sn_i^{2+}/Pb_i^{2+} + 2e^-$, thus presenting n-type conduction characteristics[26,27]. Specifically, the Seebeck coefficient of Cl doped crystals has relatively low value compared with undoped $PbSnS_2$ crystal, which is mainly attributed to the significant increase of carrier concentration.

In order to unveil the conduction behavior of Cl doped crystals, we conducted the high-temperature Hall measurement in the temperature range of 300–773 K (Fig. 2c). In most uniformly doped semiconductors, the carrier concentration is fixed by the foreign dopant and is independent of temperature[25]. However, our results show that the carrier concentration has an obvious decreasing trend with temperature rising, indicating that the cation vacancies may be excited or electrons may be redistributed among the multiple conduction bands at higher temperatures[46]. To verify this, we compare the temperature-dependent behavior of Hall carrier mobility $\mu_H$ and weighted mobility $\mu_W$ (a mobility includes the effective mass $m^*$ of electronic density of states, and is widely used in thermoelectric systems to evaluate the carrier transport behavior)[3,38,47,48], as shown in Fig. 2d. Based on the Drude-Sommerfeld free electron model $\sigma = n_H e \mu_H$ (where $e$ is electron charge), the Hall carrier mobility decreases following the law of $\sim T^{-1.5}$ with rising temperature, demonstrating that the carriers are mainly scattered by acoustic phonons[49]. On the contrary, according to the temperature-dependent electrical conductivity and Seebeck coefficient, the weighted mobility decreases following the law of $\sim T^{-1.5}$ at a relatively lower temperature range ($\sim$300–473 K), while completely deviates from this law at higher temperatures ($\sim$473–823 K) for Cl doped crystals. Namely, the calculated weighted mobility is much higher than the theoretical $T^{-1.5}$ line. It is well known that the weighted mobility is related to the Hall carrier mobility by $\mu_W \approx (m^*/m_e)^{1.5} \mu_H$ (where $m_e$ is the electron mass)[47]. Thus, density of states effective mass $m^*$ is effectively improved at high temperatures, which is contributed from the multiple band transport as discussed later. Combined with the promoted carrier concentration and electrical conductivity by Cl doping and favorable Seebeck coefficient caused by larger effective mass at high temperatures, the power factor of $PbSnS_2$ crystal maintain a relatively high value over the whole temperature range (Fig. 2e). Particularly, the values of power factor for the optimal Cl doped $PbSnS_2$ crystal at 300 K and 773 K are $\sim$6.25 and $\sim$4.26 $\mu W\ cm^{-1}\ K^{-2}$, respectively. Herein, we also investigated the thermoelectric transport performance of $PbSnS_2$ crystals along the in-plane direction and compared with that along the out-of-plane direction (Supplementary Fig. 4–6). It can be seen that the electrical transport properties exhibit strong anisotropy and the power factor values along the out-of-plane direction are superior at relatively higher temperatures, which can be attributed to the increasing out-of-plane charge density with temperature (Fig. 2e and Fig. 3). A clear increment of the charge density within Pb-S plane from 300 K to 773 K can be seen, while the charge density in the Sn-S plane undergoes a first increase and then

decrease, which is results from the temperature-dependent interplay of the conduction bands as discussed below.

Based on the high temperature synchrotron radiation X-ray diffraction (SR-XRD) data of the optimal Cl doped sample (Supplementary Fig. 7), Rietveld refinement was conducted to obtain the temperature-dependent crystal structure and atomic positions (Supplementary Fig. 8 and Supplementary Table 1-2). The density functional theory (DFT) calculations were performed to obtain the electronic band structure of Cl doped $PbSnS_2$ crystals with rising temperature (Fig. 4 and Supplementary Fig. 9). Figure 4a shows the electronic band structure of the optimal Cl doped $PbSnS_2$ crystal at 300 K. The valence band maximum (VBM) and the conduction band minimum (CBM) of the optimal Cl doped sample are located at Y point and Γ-X direction, respectively. Therefore, $PbSnS_2$ is an indirect band gap semiconductor with a band-gap of ~0.9 eV, which is in good agreement with previous calculations[44] and our experimental values (Supplementary Fig. 10). The measured carrier concentration of ~1.7 × $10^{19}\ cm^{-3}$ indicates that only the first conduction band maximum CBM1 (located along Γ-X direction) can be activated by electron doping at room temperature. We further investigate the evolution of conduction band with the temperature. Namely, the electronic band structures at 300, 573, 623, 673 and 773 K are aligned to the CBM1 (Fig. 4b). As we can see, the second conduction band maximum CBM2 (located along Γ-Y direction) and the third conduction band maximum CBM3 (located at Γ point) converge with CBM1 and then degenerate with rising temperature. The energy offsets between the CBM1 and CBM2, CBM1 and CBM3 at the temperature range of 623–673 K can be as low as ~0.12 and ~0.05 eV, respectively. In addition, we calculated the Fermi surfaces when the three conduction bands are activated (Fig. 4c). There are two kinds of Fermi surfaces in the first Brillouin zone, the sphere-like Fermi surfaces along the directions of Γ-X and Γ-Y and the dumbbell-like Fermi surfaces at the Γ point, indicating the multiple types of pockets which are from CBM1, CBM2 and CBM3, respectively. On the other hand, single-valley effective mass $m_b^*$ of CBM1, CBM2 and CBM3 shows an overall tendency of decline with rising temperature (Fig. 4d). However, the temperature-dependent density of states effective mass $m^*$ can be estimated by the Pisarenko relationship[50]:

$$S = \frac{8\pi^2 m^* k_B^2 T}{3eh^2} \left(\frac{\pi}{3n_H}\right)^{2/3} \quad (1)$$

where $k_B$ is the Boltzmann constant and $h$ is the Planck constant (Fig. 4e). Based on the formula $m^* = N_v^{2/3} m_b^*$, the number of degenerate valleys $N_v$ increases first and then slightly decreases with temperature, corresponding to the evolution process of the three conduction bands (Fig. 4b). Furthermore, the Seebeck coefficient of Cl doped $PbSnS_2$ crystals deviates to higher values obviously at higher temperatures, confirming the existence of the multiple band conduction (Fig. 4f). Our calculation results illustrate that multiple conduction bands are activated to be involved into carrier transport at high temperatures, especially at $T > 600$ K, while it is typical single conduction band transport at 300 K for Cl doped $PbSnS_2$ crystals, which is consistent with the analysis results of Fig. 3 and similar to the reported Br-doped SnSe crystals[46].

## Thermal transport properties

The thermal transport properties of $PbSnS_2$ crystals along the out-of-plane direction and the relevant theoretical calculations are shown in Fig. 5. Figure 5a shows the temperature-dependent thermal conductivity. It is apparent that the total thermal conductivity of undoped $PbSnS_2$ increases abnormally in the medium temperature range, which might be caused by the high thermal conductivity phase PbS as a product of partial decomposition (Supplementary Fig. 7a, c). On the contrary, the lattice thermal conductivity of

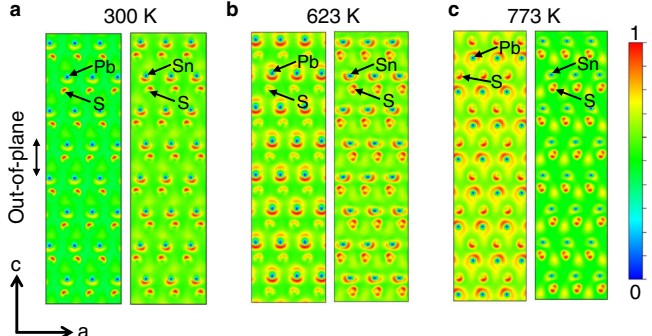

**Fig. 3 | Projected interlayer charge densities of the optimal Cl doped PbSnS₂ along [010] direction. a** 300 K. **b** 623 K. **c** 773 K.

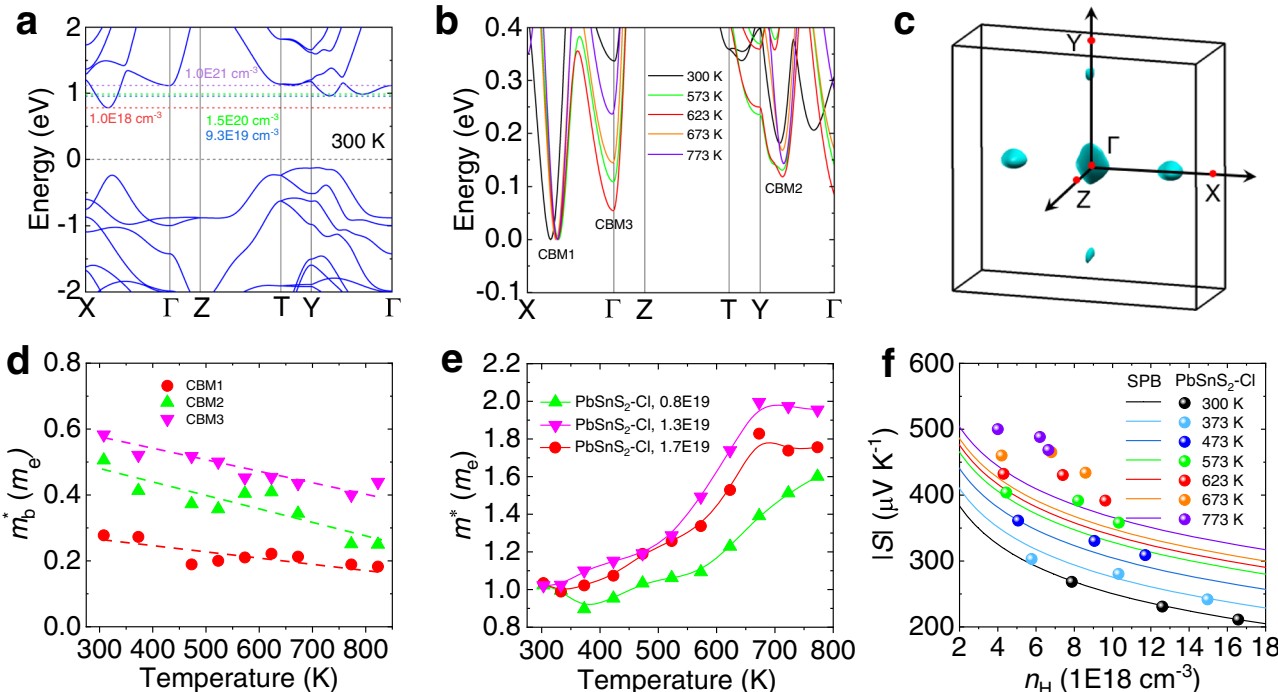

**Fig. 4 | Electronic band structures of the optimal Cl doped PbSnS₂ sample.** **a** Electronic band structure at 300 K. The black dotted line represents Fermi level, and the red, blue, green, and purple dotted lines from bottom to up represent Fermi levels with carrier concentrations of $1.0 \times 10^{18}$, $9.3 \times 10^{19}$, $1.5 \times 10^{20}$, and $1.0 \times 10^{21}\,cm^{-3}$, respectively. **b** The dynamic evolution of three separated bands (CBM1, CBM2 and CBM3) with rising temperature and the electronic band structure aligned to the CBM1. **c** Fermi surfaces when the three conductions bands are activated. **d** Temperature-dependent single-band effective mass for CBM1, CBM2 and CBM3. **e** Temperature-dependent effective mass $m^*$ for Cl doped PbSnS₂ crystals. **f** Pisarenko relationship at different temperatures.

doped PbSnS₂ crystals follows the Umklapp process in the temperature range of 300–773 K, which is due to the fact that Cl doping reduces the formation energy of PbSnS₂ and thus enhances the phase stability (Supplementary Fig. 7b, d)[42]. The lattice thermal conductivity of Cl doped PbSnS₂ crystal increases suddenly in the temperature range of 773-823 K, which is attributed to the phase transition from low symmetric *Pnma* phase to high symmetric *Cmcm* phase. It is noteworthy that the total thermal conductivity of undoped PbSnS₂ crystal at room temperature is as low as ~0.65 W m⁻¹ K⁻¹. Furthermore, the optimal Cl doped crystal has lower lattice thermal conductivity of ~0.50 W m⁻¹ K⁻¹ and ~0.25 W m⁻¹ K⁻¹ at 300 K and 773 K due to the induced point defect scattering and purer phase caused by Cl doping. The thermal diffusivity (*D*), heat capacity (*C*ₚ), Lorenz number (*L*), and electronic thermal conductivity (*κ*ₑₗₑ), and sample density (*ρ*) were presented in Supplementary Fig. 11,12,14 and Supplementary Table 3. Surprisingly, the lattice thermal conductivity of both PbSnS₂ polycrystal[35] and crystal maintain ultralow values over the whole temperature range comparing with other n-type lead and tin chalcogenides, including Cl doped PbS polycrystal[45], Br doped SnS crystal[23], and Br or Cl doped SnSe crystals[9,38] (Fig. 5b).

In order to uncover the origin of the intrinsic low lattice thermal conductivity of PbSnS₂, the phonon spectrum, Grüneisen parameters and the electron localization function (ELF) were calculated based on DFT calculations. Figure 5c shows that the maximum frequency of the acoustic phonon mode along the Γ-Z direction (corresponds to the out-of-plane direction) is ~0.57 THz, much lower than those of other directions, which may be attributed to the weak Van der Waals interaction of interlayers. In addition, the average acoustic phonon modes of PbSnS₂ along Γ-X, Γ-Y and Γ-Z directions are the softest among the typical lead and tin chalcogenides like PbS, SnS and SnSe (Fig. 5d and Supplementary Fig. 13). Such soft acoustic phonon modes indicate lower Debye temperatures and smaller phonon group velocities, which are responsible for the ultralow lattice

thermal conductivity in PbSnS₂[51–53]. Similar to the isostructural SnSe[32] and SnS[30], the unbalanced chemical bonding for Pb- and Sn-centered polyhedra formed in PbSnS₂, and strong anharmonicity can be anticipated. To evaluate the anharmonicity of PbSnS₂, the Grüneisen parameters were calculated, which represent the relationship between phonon vibration frequency and crystal volume change. Figure 5e shows the Grüneisen dispersions along highly symmetrical directions. The average Grüneisen parameters along *a*, *b* and *c* directions are shown in the inserted table, which are 2.1, 1.9 and 5.1, respectively. The unusual average Grüneisen parameters along the *c* direction is much higher than the values reported in PbS, SnS and SnSe[30,32,41], hinting a potential strong anharmonicity. Furthermore, our calculations show that there is a "mushroom" shape electron localization function (ELF) around the Sn and Pb atoms (the isosurface level is set as 0.95), which is an obvious sign of the presence of lone pair electrons (Fig. 5f). The stereochemically expressed 5 *s* or 6 *s* lone pair electrons by Sn²⁺ or Pb²⁺ could interact with neighboring Sn-S or Pb-S bonds. And the produced nonlinear repulsive electrostatic force reduces the lattice symmetry and disturbs lattice vibration[51,54], leading to the strong anharmonicity, which is similar to those observed in BiSbSe₃[55] and CuBiS₂[56]. In summary, we attribute the ultralow lattice thermal conductivity of PbSnS₂ along the out-of-plane direction to the strong anharmonicity induced by weak interaction of interlayers and sterically accommodated lone pair electrons of Sn²⁺ or Pb²⁺.

### ZT values and single-leg power generation efficiency

Combined with the tremendously improved electrical transport performance and intrinsic low lattice thermal conductivity in Cl doped PbSnS₂, we obtained advantageous *ZT* values over the whole temperature range along the out-of-plane direction (Fig. 6a). It can be seen that the maximum *ZT* (*ZT*ₘₐₓ) value of ~0.3 and ~1.2 can be achieved in the optimal Cl doped crystal at 300 and 773 K, respectively, along with

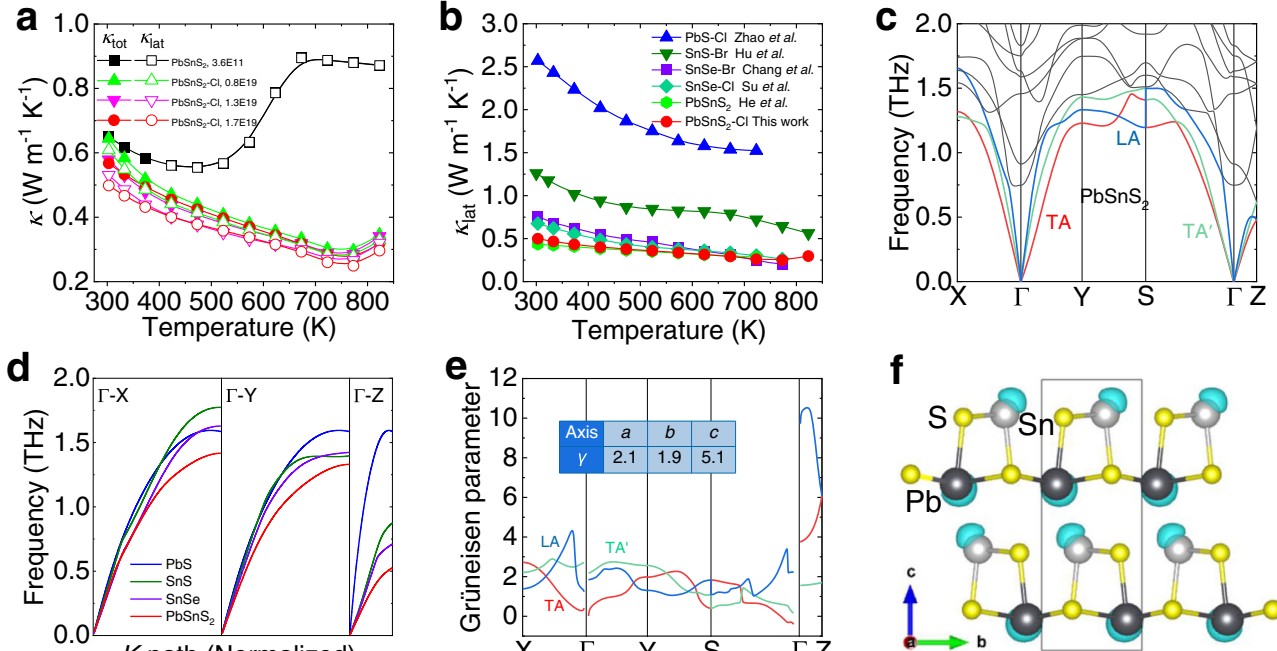

**Fig. 5 | Temperature-dependent thermal transport performance of undoped and Cl doped PbSnS₂ along the out-of-plane direction and theoretical calculation results. a** Thermal conductivity. **b** Lattice thermal conductivity comparisons of n-type lead and tin chalcogenides, including Cl doped PbS polycrystal[45], Br-doped SnS crystal[23], Br and Cl doped SnSe crystals[9,38], and PbSnS₂ polycrystal[35] and crystal. **c** Phonon dispersions spectrum. **d** Averaged acoustic mode comparison of typical lead and tin chalcogenides. **e** Grüneisen parameters, the inserted table is the average Grüneisen parameters along *a*, *b* and *c* directions. **f** Calculated ELF for PbSnS₂, where the isosurface level is set as 0.95.

a high average $ZT$ ($ZT_{ave}$) value of ~0.75 within 300–773 K (Supplementary Fig. 15), while the $ZT_{max}$ of undoped PbSnS₂ crystal is only ~0.05 at 823 K. Considering that there is a steady-state elevation in heat capacity $C_p$ near the temperature region of the structural phase transition[57,58], we compared the $C_p$ values and the final $ZT$ values of all samples according to Dulong-Petit law, Debye model and the experimental DSC data (Supplementary Fig. 14 and inset of Fig. 6a). And it is apparent that the results of these three data processing methods are consistent within the 20% error range. Figure 6b illustrates that $ZT$ values along the out-of-plane direction is superior to that along the in-plane direction, which can be attributed to the ultralow lattice thermal conductivity coming from strong interlayer phonon scattering and favorable power factor due to increased interlayer charge density with rising temperature, similar to the 3D charge and 2D phonon transport characteristics in n-type SnSe[9] (Supplementary Fig. 4–6). Furthermore, it is apparent that the thermoelectric transport performance of Cl doped PbSnS₂ outperforms that of Br-doped SnS crystals over the whole temperature range due to the much higher carrier concentration in PbSnS₂ crystals[23,24]. The high thermoelectric transport performance of n-type PbSnS₂ crystal indicate that it is expected to be a counterpart of p-type SnS crystal to fabricate low-cost and highly efficient thermoelectric devices[16].

To investigate the power generation potential of high-performance PbSnS₂ crystals experimentally, the single-leg power generation test was performed using mini-PEM equipment, as shown in the inset of Fig. 6c. The single-leg device produces a maximum output power of ~18 mW and power generation efficiency of ~2.7% with $T_c =$ 295 K and $T_h = 672$ K (Fig. 6c, d), which is comparable to the maximum conversion efficiency of ~3% achieved in p-type SnS crystals[16] (the inset of Fig. 6d). And higher energy conversion efficiency can be expected in PbSnS₂-based crystals through further optimizing the interface structure and conducting layers. Furthermore, the enormous potential for PbSnS₂ crystal could make it competitive for practical applications in thermoelectric field.

## Discussion

In this study, we discovered a promising n-type sulfide compound, PbSnS₂, and prepared its crystals successfully using a modified temperature gradient method and optimized the thermoelectric performance with effective Cl doping, which offers an eligible n-type counterpart of high-performance p-type SnS to fabricated the earth-abundant thermoelectric devices. Combined the experimental data with theoretical calculations, the thermoelectric transport properties of PbSnS₂ crystals were analyzed in detail. Firstly, superior thermoelectric performance was achieved along the out-of-plane direction due to the ultralow thermal conductivity coming from strong interlayer phonon scattering and moderate power factor coming from increased interlayer electron charge density with temperature rising, indicating the characteristic of 3D charge and 2D phonon transports for Cl doped PbSnS₂ crystals. Secondly, the carrier concentrations are boosted by Cl doping and the electrical conductivity is thus greatly improved. The activated multiple conduction bands transport at high temperature range maintains the Seebeck coefficient at a high level, which can be verified by the weighted mobility and electronic band structure calculations. Thirdly, the intrinsic ultralow lattice thermal conductivity caused by strong anharmonicity is beneficial for achieving high thermoelectric performance. Finally, we acquired a $ZT_{max}$ of ~1.2 at 773 K and $ZT_{ave}$ of ~0.75 within 300–773 K for Cl doped PbSnS₂ crystals, which is matchable to the optimized p-type SnS crystals[16]. Moreover, a potential experimental output power of ~18 mW and energy conversion efficiency of ~ 2.7% can be obtained with $T_c = 295$ K and $T_h = 672$ K in a single-leg device. Our results show that PbSnS₂ is a promising mid-temperature thermoelectric candidate, which lays a foundation for the realization of the low-cost and high-effective thermoelectric devices based on the sulfide compounds such as n-type PbSnS₂ and p-type SnS.

High-ranged thermoelectric performance can be achieved in PbSnS₂ crystals solely by electron doping. The next optimization

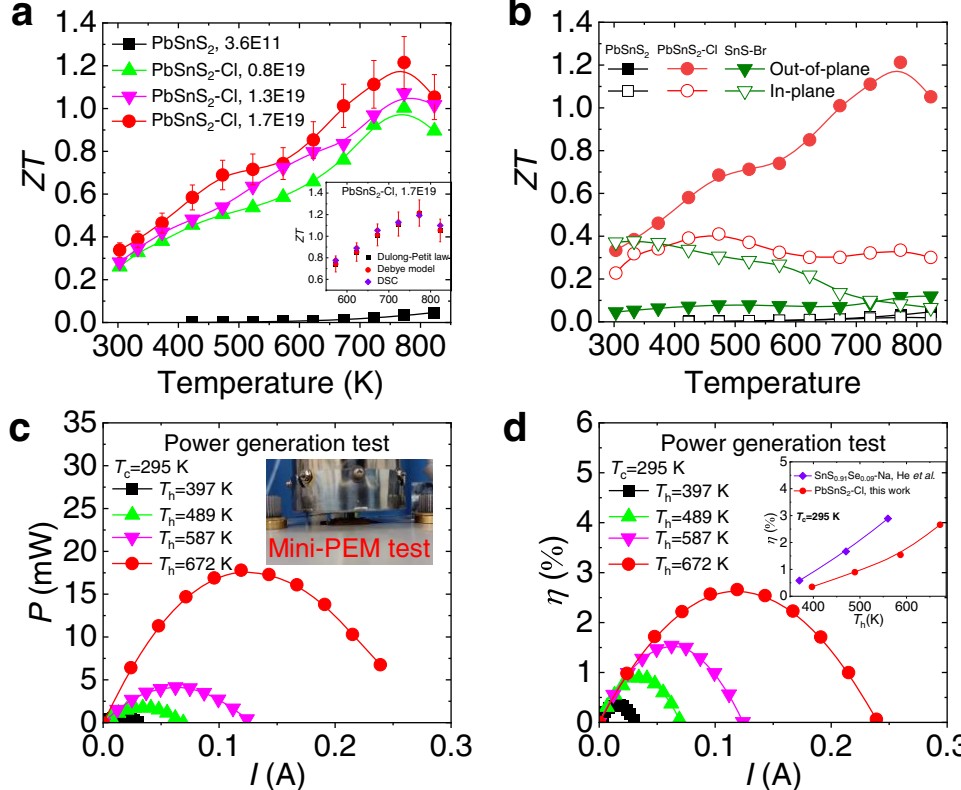

**Fig. 6 | Thermoelectric performance and single-leg power generation efficiency. a** $ZT$ values of undoped and Cl doped $PbSnS_2$ crystals, the inset shows $ZT$ values of the optimal Cl doped $PbSnS_2$ crystal calculated according to Dulong-Petit law, Debye model and the experimental DSC data. Error bars are ±10%. **b** The comparison of $ZT$ values between $PbSnS_2$ and SnS crystals[23,24] along in-plane and out-of-plane directions. **c** Output power $P$, the inset shows the mini-PEM test. **d** Power generation efficiency $\eta$ for single-leg device based on the optimal Cl doped $PbSnS_2$ crystal. Comparison of single-leg power generation efficiency between the optimal Cl doped $PbSnS_2$ crystal in this work and and p-type SnS crystal[16].

schemes can be further improving the carrier concentration of Cl doped $PbSnS_2$ by co-doping high valent elements such as $Bi^{3+}$, $Sb^{3+}$, $In^{3+}$ or $Nb^{5+}$ at the cation positions; manipulating the conduction band structure and balancing the relationship between carrier mobility and effective mass through alloying $Se^{2-}$ or $Te^{2-}$ at the anion position; increasing the charge density and strengthening phonon scattering along the out-of-plane direction by intercalating small metal atoms such as Co, Ni, Cu and Zn. These strategies for optimizing electron or phonon transport merit collaborative use to optimize thermoelectric performance of $PbSnS_2$ and can be extended to similar thermoelectric systems.

## Methods
### Sample synthesis
High-purity raw materials Pb block (99.99%, Aladdin), Sn particles (99.999%, Aladdin), S powder (99.999%, Aladdin), and $PbCl_2$ powder (99.999%, Aladdin) were weighted according to the nominal compositions of $PbSnS_{2-x}Cl_x$ ($x$ = 0, 0.02, 0.04, 0.06) and were flame-sealed into quartz tubes after vacuum. In order to prevent the sample from oxidizing due to quartz tubes rupture during the phase transformation, outer glass quartz tubes were used. These prepared samples were slowly heated up to 873 K for 10 h and then soaked at this temperature for 12 h, next heated up to 1223 K for 10 h and then soaked at this temperature for 6 h in muffle furnace. Subsequently, then polycrystalline ingots would be obtained after cooling in the furnace. The obtained ingots of ~15 g were ground into powder and put into special quartz tubes with taper. After being vacuumized and flame-sealed, the reprocessed ingots were heated up to 1313 K for 12 h and then soaked at this temperature for 10 h, next slowly cooled to 973 K at a rate of 1 K h$^{-1}$ in a vertical furnace with a temperature gradient. Subsequently,

$PbSnS_2$ crystals with size of ~$\Phi$11 × 40 mm³ were obtained after cooling in the furnace.

### Electrical transport property measurements
The obtained crystals were cut and polished into rectangles with size of ~ 3 × 3 × 7 mm³ along the in-plane and out-of-plane directions. The Seebeck coefficient and conductivity of the samples could be obtained with Ulvac Riko ZEM-3 instrument simultaneously in a thin helium atmosphere. It should be noted that the sample surface is coated with a thin layer of boron nitride to prevent contamination of the instrument by possible volatilization, and that the uncertainty of Seebeck coefficient and conductivity measurements is within 5%.

### Thermal transport property measurements
The obtained crystals were cut and polished into slices with size of ~$\Phi$6 × 1.5 mm³ along the same direction as the electrical transport performance measurements. The samples were coated with a thin layer of graphite to reduce the error caused by the emissivity of the material. The total thermal conductivity can be calculated via $\kappa = DC_p\rho$, where $D$ is the thermal diffusion coefficient, which could be obtained using Netzsch LFA 457 and analyzed using the Cowan model with pulse correction. $C_p$ represents specific heat capacity, which can be calculated according to Debye model and we have compared the $C_p$ values near the temperature region of the structural phase transition based on Dulong-Petit law, Debye model and the experimental DSC data. $\rho$ represents the density, which can be inferred from the dimensions and mass of the samples. The uncertainty of the total thermal conductivity is within 15% due to errors in measurement or calculation of $D$, $C_P$, and $\rho$. Therefore, the uncertainty of $ZT$ values obtained by the final calculation is within 20%.

## Microstructure characterization

Scanning transmission electron microscopy (STEM) studies were conducted using a Cs-corrector JEM-ARM200F NEOARM with a cold FEG at 200 kV. The thin specimens were prepared by conventional standard methods, that is, cutting, grinding, dimpling, polishing and Ar-ion milling on a liquid nitrogen cooling stage.

## High temperature synchrotron radiation X-ray diffraction (SR-XRD)

Samples of undoped $PbSnS_2$ and the optimal Cl doped $PbSnS_2$ crystal were ground into powder in an $N_2$-filled glove box, and then passed through a 400-mesh sieve before being loaded into cylindrical quartz capillary tubes with a diameter of ~0.5 mm and flame-sealed. The temperature-dependent phase structures of the samples were obtained at BL14B1 of Shanghai Synchrotron Radiation Facility (SSRF) using X-ray with a wavelength of 0.6887 Å. The samples were heated from 300 K to 823 K at a rate of 10 K/min during the measurement and GSAS software was adopted to refine the phase structures with various temperature.

## Single-leg power generation efficiency test

The single-leg device was fabricated using the optimal Cl doped $PbSnS_2$ crystal with the geometrical dimension of ~2 mm (length) × 3 mm (width) × 6 mm (height), where nickel was electroplated onto the cleavage surface as a barrier layer and gold foil was used as contact material. Mini-PEM Ulvac-Riko (Japan)[59] was adopted for the direct data of output power and conversion efficiency while the cold-side temperature ($T_c$) was maintained at 295 K and the hot-side temperature ($T_h$) was varied from 397 K to 672 K.

## Data availability

The authors declare that the data supporting the findings of this study are available on reasonable request.

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

## Acknowledgements

This work was supported by the National Key Research and Development Program of China (2018YFA0702100), the National Natural Science Foundation of China (51571007, 51772012, 52002011, 52002042, and 12204156), the Basic Science Center Project of National Natural Science Foundation of China (51788104), Beijing Natural Science Foundation (JQ18004), 111 Project (B17002) and the National Science Fund for Distinguished Young Scholars (51925101). L.-D.Z. acknowledges the support from the high performance computing (HPC) resources at Beihang University, BL14B1 at Shanghai synchrotron radiation facility (SSRF) for the SR-XRD measurements, BL01B1 at Spring-8 for the XAFS experiments (Proposal Number: 2021B1109) and Center for High Pressure Science and Technology Advanced Research (HPSTAR) for STEM measurements.

## Author contributions

L.-D.Z., D.W. and L.Z. conceived the idea, designed the experiments and supervised the research. S.Z. performed the sample synthesis, structural characterization and thermoelectric transport property measurements. Q.Z., D.W. and X.F. carried out the theoretical calculation. X.G. and T.H. performed microstructure characterization of the samples. Y.Z. conducted the X-ray absorption fine structure (XAFS) spectroscopy measurements and analyzed the data. Z.-H. G. and H. L. prepared the single-leg device and carried out the power generation text. L.S., B.Q., H.S. and S.Z. carried out the high temperature synchrotron radiation X-ray diffraction (SR-XRD) measurements and analyzed the data. S.Z., D.W., and L.-D.Z. wrote the manuscript with contributions from other authors. All authors analyzed the results and commented the manuscript.

## Competing interests

The authors declare no competing interests.
