## [Peer Review File · Nature Communications]

Realizing high-ranged thermoelectric performance in PbSnS₂ crystalsREVIEWER COMMENTS

Reviewer #1 (Remarks to the Author):

Comments: This manuscript reports on the discovery of high performance thermoelectric PbSnS₂ crystal doped by Cl with a peak zT value of 1.2. Essentially, the exploration of novel n-type sulfides is intriguing for thermoelectric community. In this work, the authors claimed the layered PbSnS₂ has great potential for application and ascribed its high performance to intrinsic low lattice thermal conductivity and multiple conduction band transport behavior at high temperature. However, this manuscript does not have enough novelty for NC and suffers from some drawbacks and cannot be accepted for publication in the present form. The main issues are:

1. The authors mentioned the potential phase decomposition and structural phase transition in the main text but the most commonly used techniques such DSC and TGA were not used for characterization. The authors must perform such measurement to accurately determine the critical temperature points for all samples including the pristine one. Moreover, the reason for the enhanced phase stability of Cl doped samples must be clarified.
2. The authors employed the Debye model for heat capacity estimation in the whole temperature. However, for the temperature region near structural phase transition, such approximation is not valid and the experimental DSC data should be adopted to make correction according to some important references (Phys. Status Solidi RRL 2016, 10, 618; Adv. Mater. 2019, 31, 1806518).
3. Both the band calculations and electrical transport data fitting indicated that a multiple conduction band transport occurs at high temperature. The attained power factor about $4\mu\text{W cm}^{-1}\text{K}^{-2}$ in this work is rather small, even inferior to most of thermoelectric compounds with single band transport feature. So the authors must provide detailed explanation for such contradiction.
4. It is claimed that the power factor values along the out-of-plane direction is superior to that along the in-plane direction. However, it is controversial to see in Figure 2f that the power factor values along the out-of-plane direction is evidently lower than that along the in-plane direction below 550 K. What is the reason?
5. If the structural phase transition indeed occurs above 773 K, the calculation of average ZT within 300-823 K is meaningless since structural phase transition leads to failure of thermoelectric device and thus such estimation is not practical.
6. The measured device efficiency of 0.8% is too low to adequately demonstrate the practical potential for this compound even though the authors refer to contact resistance. The authors must make some good devices and measure their performance if they really want to prove the practical potential of these crystals in the high profile journal like nature communications.
7. In Fig. 4 and Fig. 5, it is apparent that the high symmetry points used to describe band path in the electronic and phonon structures are not consistent. The authors should give their specific reason for this.

Reviewer #2 (Remarks to the Author):

High TE performance was found in PbSnS₂ crystals doped by Cl on S sites. Both increased electron concentration and the convergence of conduction bands were found to be responsible to the enhanced electrical transport properties. It is an interesting result. Here

are some comments:

- 1) For the single-leg test, the conversion efficiency is kind of low. Although the high contact resistance was claimed to be the reason for low efficiency, could the authors provide more detailed information on this?
- 2) Considering that Pb may be regarded as one of the main components, the "eco-friendly" may not be a proper word for this material.
- 3) Since the undoped sample decomposed at 623K, why the authors can still obtain single crystal form of this sample? I think the growth temperature is much higher than 623K, so that one would not get pure phase for undoped sample.
- 4) Why the electrical conductivity of Cl0.06 sample is lower than that of Cl0.04 sample? More Cl dopant would lead to higher electron concentration, isn't it?
- 5) For the larger out-of-plane lattice parameter, the authors claimed that it comes from interstitials Sn_i and Pb_i. However, it may just be because of the substitution of Sn atoms by larger Pb atoms.
- 6) For the decreasing electron concentration with increasing temperature, the authors claimed that it is caused by the redistribution of electrons in different conduction bands. However, if some acceptors get ionized with increasing temperature, one would also get decreasing electron concentration with increasing temperature. Considering that the cation vacancies are commonly observed in the IV-VI systems, this may be a possible reason for the decreasing electron concentration with increasing temperature.

Reviewer #3 (Remarks to the Author):

The development of low cost, high performance and homojunction structures is particularly important for the large-scale application of thermoelectric devices. Increasing progress on the thermoelectric performance improvement has been made in p-type SnS recently, while the development of n-type SnS seems to be rare. This work developed a novel n-type Sn-based sulfide PbSnS₂ through alloying Pb at Sn sites and realizing high-ranged thermoelectric performance after growing single crystals and doping Cl in it. Pb alloying was beneficial to suppress Sn vacancies and generate electrons when forming interstitials, while Cl doping enables phase stability and promotes band convergence through optimizing carrier concentration. Combining giant phonon anharmonicity along out-of-plane direction, a favorable average ZT was achieved over a wide temperature range.

This manuscript demonstrates for the first time to comprehensively introduce the PbSnS₂ as a promising n-type thermoelectric material, which is intriguing. The manuscript should get wide attentions and it will be appreciated by the broader audience. The paper is well organized and carefully written. The novelty and significance are high. Therefore, I believe that this manuscript should be accepted for publication in Nature Communications after some minor revision. Please find below few minor issues-

1. The X-ray diffraction of crystal cleavage planes prove the high quality of the synthesized crystals, and the authors should add the X-ray diffraction measurements of corresponding powder to prove that the PbSnS₂ single phase is indeed synthesized.
2. In Supplementary Fig. 2b, simulated diffraction peaks and Bragg's positions for four models of PbSnS₂ have been named repeatedly, the authors should distinguish them.
3. One key to improve the single-leg power generation efficiency is optimizing the interface structure and conducting layers. So the authors should point out which process was used in

the efficiency test.

4. As stated in the supplementary information, the uncertainty of final ZT values is within 20%. Please add error bars to the ZT curve in Fig. 6a.

5. High-ranged thermoelectric performance was realized in PbSnS₂ crystals. However, doping is only the first step to optimize the performance of this novel thermoelectric material. The authors would better provide the further potential optimization strategies for readers' reference as few lines outlook in conclusion.

Reviewer #1 (Remarks to the Author):

Comments: This manuscript reports on the discovery of high performance thermoelectric PbSnS₂ crystal doped by Cl with a peak ZT value of 1.2. Essentially, the exploration of novel n -type sulfides is intriguing for thermoelectric community. In this work, the authors claimed the layered PbSnS₂ has great potential for application and ascribed its high performance to intrinsic low lattice thermal conductivity and multiple conduction band transport behavior at high temperature. However, this manuscript does not have enough novelty for NC and suffers from some drawbacks and cannot be accepted for publication in the present form. The main issues are:

Response: We appreciate your valuable comments. And we carefully revised the paper. Hopefully, our revised manuscript could meet your expectations.

1. The authors mentioned the potential phase decomposition and structural phase transition in the main text but the most commonly used techniques such DSC and TGA were not used for characterization. The authors must perform such measurement to accurately determine the critical temperature points for all samples including the pristine one. Moreover, the reason for the enhanced phase stability of Cl doped samples must be clarified.

Response: Thanks for your valuable comment. We have conducted DSC test to confirm the partial decomposition temperature of the pure sample and structural phase transition temperature of all samples. The results show that the partial decomposition temperature of the pure sample is ~ 623 K, and the structural phase transition temperature of all samples are in the temperature range of 773-823 K. And the results are consistent with the high temperature synchrotron radiation X-ray diffraction (SR-XRD) data. As for the phase stability, we further conducted theoretical calculations on the formation energy of the compound with different compositions, as shown in Supplementary Fig. 7d. The formation energy decreases with the increasing of Cl doping content in PbSnS₂, which may be the reason for the enhanced phase stability of the Cl doped samples. Considering the different doping efficiency of Cl element in doped samples, we named the samples based on the room-temperature carrier concentration in our revised manuscript.

Revision: It is apparent that the total thermal conductivity of undoped PbSnS₂ increases abnormally in the medium temperature range, which may be caused by the high thermal conductivity phase PbS as product of partial decomposition (Supplementary Fig. 7a, c). On the contrary, the lattice thermal conductivity of doped PbSnS₂ crystals follows the Umklapp process in the temperature range of 300-773 K, which is due to the fact that Cl doping reduces the formation energy of PbSnS₂ and thus enhances the phase stability (Supplementary Fig. 7b, d).

Supplementary Figure 7. High temperature SR-XRD data of (a) undoped PbSnS_2 and (b) the optimal Cl doped PbSnS_2 . (c) DSC measurements of Cl doped PbSnS_2 . The partial decomposition temperature of the undoped sample is ~ 623 K, and the structural phase transition temperature of all samples are in the temperature range of 773-823 K, which are consistent with the SR-XRD data. (d) The comparison of the formation energy between undoped and Cl doped PbSnS_2 with different compositions. Phase stability of PbSnS_2 was enhanced due to lower formation energy after Cl doping.

2. The authors employed the Debye model for heat capacity estimation in the whole temperature. However, for the temperature region near structural phase transition, such approximation is not valid and the experimental DSC data should be adopted to make correction according to some important references (Phys. Status Solidi RRL 2016, 10, 618; Adv. Mater. 2019, 31, 1806518).

Response: Thanks for your valuable comment. Considering a steady-state elevation in heat capacity near the structural phase transition point (Brown D. R., et al. Phys. Status Solidi RRL 10, 618-621 (2016); Chen H., et al. Adv. Mater. 31, 1806518 (2019)), we conducted DSC test for all samples. And we compared the ZT values calculated according to Dulong-Petit law, Debye model and the experimental DSC data, which can be seen in Supplementary Fig. 14 and inset of Fig. 6a.

Revision: Considering that there is a steady-state elevation in heat capacity C_p near the temperature region of the structural phase transition, we compared the C_p values and the final ZT values of all samples according to Dulong-Petit law, Debye model and the experimental DSC data (Supplementary Fig. 14 and inset of Fig. 6a). And it is apparent that the results of these three data processing methods are consistent within the 20% error range.

Fig. 6a ZT values of undoped and Cl doped PbSnS_2 crystals, the inset shows ZT values of the optimal Cl doped PbSnS_2 crystals calculated according to Dulong-Petit law, Debye model and the experimental DSC data.

Supplementary Figure 14. (a, d, g and j) Temperature-dependent heat capacity of undoped and Cl doped PbSnS_2 obtained from different methods, including Dulong-Petit law, Debye model and experimental DSC data. (b, e, h and k) Temperature-dependent ZT values along in-plane direction of undoped and Cl doped PbSnS_2 calculated according to different heat capacities. (c, f, i and l) Temperature-dependent ZT values along out-of-plane direction of undoped and Cl doped PbSnS_2 calculated according to different heat capacities. The results of these three different methods showed a good consistency within the 20% error range.

3. Both the band calculations and electrical transport data fitting indicated that a multiple conduction band transport occurs at high temperature. The attained power factor about $4 \mu\text{W cm}^{-1} \text{K}^{-2}$ in this work is rather small, even inferior to most of thermoelectric compounds with single band transport feature. So the authors must provide detailed explanation for such contradiction.

Response: Thanks for your valuable comment. In this work, the high performance was obtained along the out-of-plane direction in PbSnS₂ crystals due to the ultralow thermal conductivity, however, the power factors are much lower than some thermoelectric compounds with single band transport feature.

The power factor is determined by the Seebeck coefficient and electrical conductivity, which can be further attributed to the carrier concentration, effective mass, and carrier mobility. To solve the aforementioned contradiction, we systematically compare the thermoelectric transport properties of the optimal Cl doped PbSnS₂ and *n*-type PbS. The sample PbS+0.04%PbCl₂ (Zhao L.-D., *et al. J. Am. Chem. Soc.* **133**, 20476-20487 (2011)) is selected for comparison because it has single band transport feature and similar carrier concentration ($2.75 \times 10^{19} \text{ cm}^{-3}$) with the optimal Cl doped PbSnS₂ ($1.7 \times 10^{19} \text{ cm}^{-3}$).

In the Fig. R1, the Seebeck coefficient of Cl doped PbSnS₂ is obviously higher due to the larger effective mass m^* ($m^* = 1.03 m_e$ for Cl doped PbSnS₂ and $m^* = 0.40 m_e$ for PbS+0.04%PbCl₂, m_e is the electron mass) and multiple conduction bands transport at higher temperatures. The larger effective mass leads to a lower carrier mobility in Cl doped PbSnS₂ ($\mu_H = 53 \text{ cm}^2 \text{ V}^{-1} \text{ s}^{-1}$) along the out-of-plane direction compared with PbS+0.04%PbCl₂ ($\mu_H = 288 \text{ cm}^2 \text{ V}^{-1} \text{ s}^{-1}$), which is an important part of the electrical conductivity.

Therefore, the obtained power factor about $4 \mu\text{W cm}^{-1} \text{ K}^{-2}$ in this work can be attributed to the low carrier mobility and electrical conductivity. And improving the electric transport performance of the intrinsically low thermal conductive material PbSnS₂ through enhancing its carrier mobility is exactly what we will focus on next.

Figure R1. Comparison of the thermoelectric properties between the optimal Cl doped PbSnS₂ and single band transport featured PbS+0.04%PbCl₂. (a) Electrical conductivity. (b) Seebeck coefficient. (c) Power factor. (d) Total and (e) lattice thermal conductivity. (f) ZT values.

4. It is claimed that the power factor values along the out-of-plane direction is

superior to that along the in-plane direction. However, it is controversial to see in Figure 2f that the power factor values along the out-of-plane direction is evidently lower than that along the in-plane direction below 550 K. What is the reason?

Response: We are sorry for this confusion. In particular, we refer to the power factor along the out-of-plane direction is superior to that along the in-plane direction at higher temperatures. Relevant ambiguous statements have been corrected in revised manuscript.

As can be seen in Supplementary Fig. 6, the power factor along the out-of-plane direction is lower compared with that along the in-plane direction at a relatively low temperature range (300-573 K), which can be attributed to the lower carrier mobility due to interlayer scattering basing on $\sigma = n_{\text{HE}}\mu_{\text{H}}$. However, the power factor along the out-of-plane direction is superior at relatively high temperature range due to the gradual increase of interlayer charge density (Fig. 3), which makes PbSnS₂ also possess the characteristics of 3D charge and 2D phonon transports like *n*-type SnSe and SnS (Chang C., et al. *Science* **360**, 778-783 (2018); Hu X., et al. *Scripta Mater.* **170**, 99-105 (2019)).

Revision: Compared with the in-plane direction, our results show that higher thermoelectric performance is achieved along the out-of-plane direction due to the ultralow thermal conductivity from strong interlayer phonon scattering and superior power factor at higher temperatures (>600K) due to increased interlayer charge density, which is similar to the anisotropy of *n*-type SnSe.

5. If the structural phase transition indeed occurs above 773 K, the calculation of average *ZT* within 300-823 K is meaningless since structural phase transition leads to failure of thermoelectric device and thus such estimation is not practical.

Response: Thanks for the valuable comment. During the revision stage, we calculated the average *ZT* within 300-773 K and presented in Supplementary Fig. 15.

Revision:

Supplementary Figure 15. Average ZT values for undoped PbSnS_2 ($3.6 \times 10^{11} \text{ cm}^{-3}$) and Cl doped PbSnS_2 (0.8×10^{19} , 1.3×10^{19} , $1.7 \times 10^{19} \text{ cm}^{-3}$).

6. The measured device efficiency of 0.8% is too low to adequately demonstrate the practical potential for this compound even though the authors refer to contact resistance. The authors must make some good devices and measure their performance if they really want to prove the practical potential of these crystals in the high profile journal like nature communications.

Response: Thanks for the valuable comment. The fabrication process and contacting materials of the device have great impact on the measurement of the device efficiency. Optimizing the fabrication process and contacting materials of the device can further optimize the conversion efficiency. On this basis, we have fabricated a new device using electroplated nickel as the barrier layer and gold foil as metallization contacting layer. The maximum single-leg power generation efficiency measured in PbSnS_2 is $\sim 2.7\%$ with a maximum output power of $\sim 18 \text{ mW}$ at a temperature difference of $\sim 377 \text{ K}$ (Fig. 6c,d), which is comparable to the maximum conversion efficiency of $\sim 3.0\%$ achieved in p -type SnS crystals (He W., et al. *Science* **365**, 1418-1424 (2019)). However, we must say that the current device performance is still far away from expected, but this optimization by electroplating the barrier layer has to some extent reflecting the application potential for PbSnS_2 crystals, especially considering the great difficulty of reducing the contact resistance in the out-of-plane direction of the crystals.

Revision:

Fig. 6c Output power P , the inset shows the mini-PEM test. **d** Power generation efficiency η for single-leg device based on the optimal Cl doped PbSnS_2 crystal. Comparison of single-leg power generation efficiency between the optimal Cl doped PbSnS_2 crystal in this work and p -type SnS crystal.

7. In Fig. 4 and Fig. 5, it is apparent that the high symmetry points used to describe band path in the electronic and phonon structures are not consistent. The authors

should give their specific reason for this.

Response: We are sorry for this confusion. In the electronic band structure calculation of Fig. 4, we mainly focus on the relative positions of CBM1, CBM2 and CBM3. The lattice thermal conductivity is mainly originated from acoustic phonon mode, thus the phonon spectrum presented along a , b and c directions in Fig. 5. Our different focus leads to the inconsistency of high symmetry points in the band structure and phonon spectrum calculations.

Reviewer #2 (Remarks to the Author):

Comments: High TE performance was found in PbSnS₂ crystals doped by Cl on S sites. Both increased electron concentration and the convergence of conduction bands were found to be responsible to the enhanced electrical transport properties. It is an interesting result. Here are some comments:

Response: Thanks for your affirmation of our work. Your insightful comments will help to improve our work.

1. For the single-leg test, the conversion efficiency is kind of low. Although the high contact resistance was claimed to be the reason for low efficiency, could the authors provide more detailed information on this?

Response: Thanks for your good suggestions. As for using crystal samples to fabricate thermoelectric devices, the contacting resistance is rather high due to the special and oriented arrangement of atoms, especially for the out-of-plane direction. We further optimized the fabrication process and contacting materials of the device and achieved better results. The maximum single-leg power generation efficiency after optimization can reach ~2.7%, where nickel was electroplated onto the cleavage surface as a barrier layer and gold foil was used as contact material to lower contact resistance.

Revision: Single-leg power generation efficiency test. The single-leg device was fabricated using the optimal Cl doped PbSnS₂ crystal with the geometrical dimension of ~ 2 mm (length) × 3 mm (width) × 6 mm (height), where nickel was electroplated onto the cleavage surface as a barrier layer and gold foil was used as contact material. Mini-PEM Ulvac-Riko (Japan) was adopted for the direct data of output power and conversion efficiency while the cold-side temperature (T_c) was maintained at 295 K and the hot-side temperature (T_h) was varied from 397 K to 672 K.

2. Considering that Pb may be regard as one of the main components, the "eco-friendly" may not be a proper word for this material.

Response: Thanks for your good suggestions. We have carefully examined and modified this inappropriate word in our revised manuscript.

3. Since the undoped sample decomposed at 623 K, why the authors can still obtain single crystal form of this sample? I think the growth temperature is much higher than 623K, so that one would not get pure phase for undoped sample.

Response: We are sorry for this confusion. Both the SR-XRD and DSC tests show that the undoped sample decomposed at ~ 623 K and the phase stability was enhanced due to lower formation energy after Cl doping (Supplementary Fig. 7). And the highest growth temperature of PbSnS₂ is 1313 K, which is significantly higher than

the decomposition temperature of ~ 623 K. This may be the reason why undoped crystal could be obtained. Also, similar phenomenon has been seen in SnS_2 , which decomposes at ~ 873 K (*Lide, David R. CRC Handbook of Chemistry and Physics. Boca Raton, Florida: CRC Press. 2009*) and crystals can still be prepared with highest growth temperature of ~ 1123 K (*Sharp L., et al. Cryst. Growth Des. 6, 1523-1527 (2006)*).

Supplementary Figure 7. High temperature SR-XRD data of (a) undoped PbSnS_2 and (b) the optimal Cl doped PbSnS_2 . (c) DSC measurements of undoped and Cl doped PbSnS_2 . The partial decomposition temperature of the undoped sample is ~ 623 K, and the structural phase transition temperature of all samples are in the temperature range of 773-823 K, which are consistent with the SR-XRD data. (d) The comparison of the formation energy between undoped and Cl doped PbSnS_2 with different compositions. Phase stability of PbSnS_2 was enhanced due to lower formation energy after Cl doping.

4. Why the electrical conductivity of Cl0.06 sample is lower than that of Cl0.04 sample? More Cl dopant would lead to higher electron concentration, isn't it?

Response: We are sorry for this confusion. Cl dopant is used to regulate the electron concentration of PbSnS_2 in this work. However, the solid solubility of Cl element in the matrix is limited, and it is reasonable that the electron concentration will not increase when the solid solubility is exceeded. The Cl elements that do not

incorporate into the PbSnS_2 lattice will cause many defects and impurities containing Cl, thus reducing the carrier mobility of the material and deteriorating the electric transport performance. The phenomenon that the carrier concentration decreases when the dopant is excessive slightly also appears in Na-doped SnS crystals (Wu H., *et al. Adv. Energy. Mater.* **8**, 1800087 (2018)). Moreover, considering these reasons, we named the samples based on the room-temperature carrier concentration, which was also conducted in Br and Cl doped SnSe crystals (Chang C., *et al. Science* **360**, 778-783 (2018); Su L., *et al. Science* **375**, 1385-1389 (2022)).

5. For the larger out-of-plane lattice parameter, the authors claimed that it come from interstitials Sn_i and Pb_i . However, it may just because of the substitution of Sn atoms by larger Pb atoms.

Response: Sorry for confusing you. According to existing studies in the literature, Pb substitution enlarges the interlayer distance due to its larger ionic radius, which makes the formation of Sn and Pb interstitials easier because of the reduction of the formation enthalpies of Sn and Pb interstitials (Xiao Z., *et al. Appl. Phys. Lett.* **106**, 152103 (2015)). Therefore, the logic is that the larger out-of-plane lattice parameter in PbSnS_2 compared with SnS comes from the substitution of larger Pb atoms, which might lead to interstitials Sn_i and Pb_i .

Revision: Also, the lattice parameter of PbSnS_2 along the out-of-plane direction increases by $\sim 0.24 \text{ \AA}$ compared with SnS because of the enlarged interlayer distance by larger Pb^{2+} ions substitution, which is conducive to the formation of interstitials Sn_i and Pb_i .

6. For the decreasing electron concentration with increasing temperature, the authors claimed that it is caused by the redistribution of electrons in different conduction bands. However, if some acceptors get ionized with increasing temperature, one would also get decreasing electron concentration with increasing temperature. Considering that the cation vacancies are commonly observed in the IV-VI systems, this may be a possible reason for the decreasing electron concentration with increasing temperature.

Response: Thanks for your valuable comment. This may be another reason why the carrier concentration decreases with temperature. And we included the possibility in revised manuscript.

Revision: However, our results show that the carrier concentration has an obvious decreasing trend with temperature rising, indicating that the cation vacancies may be excited or electrons may be redistributed among the multiple conduction bands at higher temperatures.

Reviewer #3 (Remarks to the Author):

Comments: The development of low cost, high performance and homojunction structures is particularly important for the large-scale application of thermoelectric devices. Increasing progress on the thermoelectric performance improvement has been made in *p*-type SnS recently, while the development of *n*-type SnS seems to be rare. This work developed a novel *n*-type Sn-based sulfide PbSnS₂ through alloying Pb at Sn sites and realizing high-ranged thermoelectric performance after growing single crystals and doping Cl in it. Pb alloying was beneficial to suppress Sn vacancies and generate electrons when forming interstitials, while Cl doping enables phase stability and promotes band convergence through optimizing carrier concentration. Combining giant phonon anharmonicity along out-of-plane direction, a favorable average *ZT* was achieved over a wide temperature range. This manuscript demonstrates for the first time to comprehensively introduce the PbSnS₂ as a promising *n*-type thermoelectric material, which is intriguing. The manuscript should get wide attentions and it will be appreciated by the broader audience. The paper is well organized and carefully written. The novelty and significance are high. Therefore, I believe that this manuscript should be accepted for publication in Nature Communications after some minor revision. Please find below few minor issues:

Response: Thanks for your positive comments, and we provided a point-to-point response as follows.

1. The X-ray diffraction of crystal cleavage planes prove the high quality of the synthesized crystals, and the authors should add the X-ray diffraction measurements of corresponding powder to prove that the PbSnS₂ single phase is indeed synthesized.

Response: Thanks for your good suggestion. We have supplemented powder X-ray diffraction measurements to show that the sample is indeed single-phase. We present this experimental result as Supplementary Fig. 1. And we updated the sequence of patterns in supplementary information due to the addition of powder X-ray diffraction patterns.

Revision: Powder X-ray diffraction patterns for undoped PbSnS₂ (carrier concentration is $3.6 \times 10^{11} \text{ cm}^{-3}$) and Cl doped PbSnS₂ (carrier concentrations are 0.8×10^{19} , 1.3×10^{19} and $1.7 \times 10^{19} \text{ cm}^{-3}$, respectively) prove that the single phases were synthesized (Supplementary Fig. 1). While the X-ray diffraction patterns for corresponding cleavage plane present only two diffraction peaks located at 31.5° and 65.5° , which can be assigned to the (004) and (008) crystal planes within the angle range of scanning, indicating the high quality of PbSnS₂ crystals (Fig. 1b).

Supplementary Figure 1. (a) Room-temperature powder X-ray diffraction patterns. (b) Refined lattice parameters along different crystal directions for undoped and Cl doped PbSnS₂.

2. In Supplementary Fig. 2b, simulated diffraction peaks and Bragg's positions for four models of PbSnS₂ have been named repeatedly, the authors should distinguish them.

Response: Thanks for the reminder. We have distinguished simulated diffraction peaks and corresponding Bragg's positions of four models through renaming them in Supplementary Fig. 3b.

Revision:

Supplementary Figure 3. Total energy and comparison of relative calculation with experimental data for four models. (a) Total energy. Comparison of simulated (b)

diffraction peaks, (c) Pb L3-edge and (d) Sn K-edge with relative experimental data.

3. One key to improve the single-leg power generation efficiency is optimizing the interface structure and conducting layers. So the authors should point out which process was used in the efficiency test.

Response: Thanks for your good suggestions. We further optimized the fabrication process and contacting material of the device and achieved better results. The maximum single-leg power generation efficiency after optimization can reach ~2.7%, where nickel was electroplated onto the cleavage surface as a barrier layer and gold foil was used as contact material to lower contact resistance.

Revision: Single-leg power generation efficiency test. The single-leg device was fabricated using the optimal Cl doped PbSnS₂ crystal with the geometrical dimension of ~ 2 mm (length) × 3 mm (width) × 6 mm (height), where nickel was electroplated onto the cleavage surface as a barrier layer and gold foil was used as contact material. Mini-PEM Ulvac-Riko (Japan) was adopted for the direct data of output power and conversion efficiency while the cold-side temperature (T_c) was maintained at 295 K and the hot-side temperature (T_h) was varied from 397 K to 672 K.

4. As stated in the supplementary information, the uncertainty of final ZT values is within 20%. Please add error bars to the ZT curve in Fig. 6a.

Response: Thanks for your good suggestion. We have added error bars to the optimal performance curve in revised manuscript considering of the final test error.

Revision:

Fig. 6a ZT values of undoped and Cl doped PbSnS_2 crystals, the inset shows ZT values of the optimal Cl doped PbSnS_2 crystals calculated according to Dulong-Petit law, Debye model and the experimental DSC data.

5. High-ranged thermoelectric performance was realized in PbSnS_2 crystals. However, doping is only the first step to optimize the performance of this novel thermoelectric material. The authors would better provide the further potential optimization strategies for readers' reference as few lines outlook in conclusion.

Response: Thanks for your good suggestion. The next optimization schemes can be further improving the carrier concentration of Cl doped PbSnS_2 by co-doping high valent elements such as Bi^{3+} , Sb^{3+} , In^{3+} or Nb^{5+} at the cation positions; adjusting the conduction band structure and balancing the relationship between carrier mobility and effective mass through alloying Se^{2+} or Te^{2+} at the anion position; increasing the charge density and strengthening phonon scattering along the out-of-plane direction by intercalating small metal atoms such as Co, Ni, Cu and Zn.

Revision: We were able to achieve high-ranged thermoelectric performance in PbSnS_2 crystals solely by electron doping. The next optimization schemes can be further improving the carrier concentration of Cl doped PbSnS_2 by co-doping high valent elements such as Bi^{3+} , Sb^{3+} , In^{3+} or Nb^{5+} at the cation positions; manipulating the conduction band structure and balancing the relationship between carrier mobility and effective mass through alloying Se^{2-} or Te^{2-} at the anion position; increasing the charge density and strengthening phonon scattering along the out-of-plane direction by intercalating small metal atoms such as Co, Ni, Cu and Zn. These strategies for optimizing electron or phonon transport merit collaborative use to optimize thermoelectric performance of PbSnS_2 and can be extended to similar thermoelectric systems.

REVIEWERS' COMMENTS

Reviewer #1 (Remarks to the Author):

Authors have successfully addressed my concerns. I suggest acceptance for publication.

Reviewer #2 (Remarks to the Author):

Most of my concerns have been addressed except comment 3. In the manuscript the authors mentioned that pure sample is not stable at high temperature and Cl-doping is helpful for the stabilization of the phase, which however, is conflict with the fact that they can obtain the pure sample in single-crystal form. Their response to this comment doesn't convince me. They provided an example that SnS₂ is not stable at high temperatures in one reference, but single crystal can be obtained in another reference. To me, the two references conflict to each other and should not be used to support the statement that unstable phase can be grown into single crystal form. Maybe the PbSnS₂ phase is stable at high temperature so that they can obtain single crystals, but somehow unstable in the test (XRD and DSC) condition. I would suggest the authors to reconsider their statement.

BTW, in the high temperature XRD patterns, a small peak can be observed at about 16 degree for the Cl-doped sample but not for the pure sample. Is it an impurity peak?

Reviewer #3 (Remarks to the Author):

Authors have satisfactorily answered the comments of all the reviewers and revised the paper accordingly.

The paper should be accepted as is.

Reviewer #1 (Remarks to the Author):

Comments: Authers have successfully addressed my concerns. I suggest acceptance for publication.

Response: Thanks for your recognition of our work.

Reviewer #2 (Remarks to the Author):

Comments: Most of my concerns have been addressed except comment 3. In the manuscript the authors mentioned that pure sample is not stable at high temperature and Cl-doping is helpful for the stabilization of the phase, which however, is conflict with the fact that they can obtain the pure sample in single-crystal form. Their response to this comment doesn't convince me. They provided an example that SnS₂ is not stable at high temperatures in one reference, but single crystal can be obtained in another reference. To me, the two references conflict to each other and should not be used to support the statement that unstable phase can be grown into single crystal form. Maybe the PbSnS₂ phase is stable at high temperature so that they can obtain single crystals, but somehow unstable in the test (XRD and DSC) condition. I would suggest the authors to reconsider their statement. BTW, in the high temperature XRD patterns, a small peak can be observed at about 16 degree for the Cl-doped sample but not for the pure sample. Is it an impurity peak?

Response: Thanks for your valuable comment and we have reconsidered our statement based on your suggestion. As shown in Supplementary Fig. 7 and Fig. 5a, the experimental facts including PbS and SnS can be indexed in undoped PbSnS₂ matrix at ~623 K, an obvious endothermic peak is found in undoped PbSnS₂ crystal at ~623K and the abnormal thermal conductivity in the medium temperature range lead us to the conclusion that undoped PbSnS₂ crystal decomposes at ~ 623 K. In addition, the thermodynamic stability calculations of Pb-Sn-S ternary system show that PbSnS₂ will decompose into SnS and PbS according to Materials Project database (Fig. R1) (<https://materialsproject.org/materials/mp-1218951?chemsys=Pb-Sn-S>; Jain A., et al. *APL Mater.* **1**, 011002 (2013); One S. P., et al. *Chem. Mater.* **20**, 1798-1807 (2008)). And the results of formation energy calculation show that Cl doping reduces the formation energy of PbSnS₂, which may be the reason for the higher stability of the Cl doped PbSnS₂ crystals. However, we obtained PbSnS₂ crystal using the modified temperature gradient method experimentally. The mechanism of the structural decomposition and crystal growth of PbSnS₂ may involve complex thermodynamic process, which need to be further studied detailly.

As shown in Supplementary Fig. 7b, small peaks which are observed at about 16° within 300-723 K for the Cl doped sample can be indexed as the PbCl₂ phase, indicating that the solid solubility of PbCl₂ in PbSnS₂ matrix is limited. And the appearance of the second phase due to exceeding the solid solubility is common in crystals (Qin B. C., et al. *J. Am. Chem. Soc.* **142**, 5901-5909 (2020)).

Supplementary Figure 7. High temperature SR-XRD data of (a) undoped PbSnS₂ and (b) the optimal Cl doped PbSnS₂. PbCl₂ phase can be indexed within 300-723 K in the optimal Cl doped PbSnS₂, indicating the exceeding solid solubility of PbCl₂ in PbSnS₂ matrix. (c) DSC measurements of undoped and Cl doped PbSnS₂. Undoped PbSnS₂ might start to partially decompose into PbS and SnS at ~ 623 K, and the structural phase transition temperature of all samples are in the temperature range of 773-823 K, which are consistent with the SR-XRD data. (d) The comparison of the formation energy between undoped and Cl doped PbSnS₂ with different compositions. Phase stability of PbSnS₂ was enhanced due to lower formation energy after Cl doping.

Fig. 5a Temperature-dependent thermal transport performance of undoped and Cl doped PbSnS_2 along the out-of-plane direction.

Fig. R1 Thermodynamic stability calculations of Pb-Sn-S ternary system according to Materials Project database (<https://materialsproject.org/materials/mp-1218951?chemsys=Pb-Sn-S>; Jain A., et al. *APL Mater.* **1**, 011002 (2013); One S. P., et al. *Chem. Mater.* **20**, 1798-1807 (2008)).

Revision: And it might be partially decomposed into PbS and SnS at ~ 623 K, while PbS as a high thermal conductivity phase is harmful to thermoelectric transport.

It is apparent that the total thermal conductivity of undoped PbSnS_2 increases abnormally in the medium temperature range, which might be caused by the high

thermal conductivity phase PbS as a product of partial decomposition (Supplementary Fig. 7a, c).

Reviewer #3 (Remarks to the Author):

Comments: Authors have satisfactorily answered the comments of all the reviewers and revised the paper accordingly. The paper should be accepted as is.

Response: Thanks for your recognition of our work.